# The Study on Mechanical Strength of Titanium-Aluminum Dissimilar Butt Joints by Laser Welding-Brazing Process

**DOI:** 10.3390/ma12050712

**Published:** 2019-02-28

**Authors:** Xiongfeng Zhou, Ji’an Duan, Fan Zhang, Shunshun Zhong

**Affiliations:** 1College of Mechanical and Electrical Engineering, Central South University, Changsha 410083, China; Xiongfeng_zhou004@126.com (X.Z.); to18373290071@163.com (S.Z.); 2State Key Laboratory of High Performance and Complex Manufacturing, Central South University, Changsha 410083, China

**Keywords:** laser welding–brazing, tensile strength, aluminum, titanium

## Abstract

Laser welding–brazing of 5A06 aluminum to Ti6Al4V titanium in a butt configuration was carried out to discuss the influences of welding parameters on dissimilar joint properties. The effects of laser offset, welding speed, and laser power on the spreading length of the molten aluminum liquid, interface fracture zone width (IFZW), fracture roughness, intermetallic compounds (IMCs) thickness, and tensile strength were also investigated. The microstructure and fracture of the joint were also studied. The results show that the tensile strength of the joint is not only influenced by the thickness and type of IMCs, but also influenced by the spreading ability of the aluminum liquid, the fracture area broken at the Ti/fusing zone (FZ) interface, and the relative area of the brittle and ductile fracture in FZ. A dissimilar butt joint with an IMC thickness of 2.79 μm was obtained by adjusting the laser offset, welding speed, and laser power to 500 μm, 11 mm/s and 1130 W, respectively. The maximum tensile strength of the joint was up to 183 MPa, which is equivalent to 83% of the tensile strength of the 5A06 aluminum alloy.

## 1. Introduction

In order to meet environmental and economic needs, modern industry has put forward higher requirements for new processes and structures [1]. The combination of multiple lightweight materials can give full play to the main advantages of each component, which not only improves the utilization rate of materials but also reduces the weight of the whole structure. Because of the low density, low cost, good forming property of the aluminum alloy [2], the excellent erosion resistance and biocompatibility, high temperature resistance, and the strength of the titanium alloy, titanium–aluminum hybrid structures have a wide range of potential applications in automotive, shipbuilding, and aerospace industries [3], which can significantly reduce weight and improve combination properties of the structure, such as corrosion resistance and high temperature resistance [4]. For example, titanium–aluminum hybrid structures have been applied to the honeycomb sandwich structure and seat-track of aircraft and have obtained good application results. However, due to the significant difference in chemical and physical properties, it is difficult for the aluminum alloy and titanium alloy to be perfectly welded together. In addition, brittle intermetallic compounds (IMCs) such as Ti_3_Al, TiAl, TiAl_2_, and TiAl_3_ are easily formed during the welding process [5]. These compounds have serious adverse effects on the weld quality of the joint because of their high brittleness and hardness. Successful welding of the aluminum alloy and titanium alloy plates requires not only overcoming a series of problems caused by performance differences, such as hot cracking, metallurgical incompatibility, and element burning, but also minimizing the thickness of brittle IMCs [6].

In recent years, many researchers have tried to use brazing, diffusion welding [7,8], explosive welding [9,10,11,12], friction welding [13], and friction stir welding [14,15,16,17] to connect titanium alloy and aluminum alloy. However, these solid state welding methods are usually limited by the structure size and shape configuration of the joints, so only simple geometric shapes can be welded, such as overlap and butt joints [18,19]. Compared with other welding methods, laser welding is widely used in welding of dissimilar metals because of the high thermal density and heat concentration, fast processing speed, small thermal deformation, and low pollution [20,21,22,23,24]. The feasibility of laser welding of titanium and aluminum has been reported via various auxiliary processes. As studied by Casalino et al. [25], when the laser offset was on the side of the titanium alloy, the high strain was located in the AA6061 aluminum alloy side, while the Ti6Al4V titanium alloy side was almost not deformed, and the rupture occurred in the fused zone that was close to the titanium alloy side. Chen et al. [26] demonstrated the feasibility of laser penetration welding of a titanium alloy and an aluminum alloy in a lap joint configuration mode. The weld geometry and the formation of IMCs could be regulated by process parameters. Leo et al. [27] studied the laser welding of a Ti6Al4V titanium alloy and an AA5754 aluminum alloy and reported the influence of post-welding heat treatments on the microstructure of the weld seam. Tomashchuk et al. [28] carried out the laser welding of a T40 titanium alloy and an AA5754 aluminum alloy with use of Al–Si filling material. They demonstrated that the important factors that related to the strength of the joint in a V groove configuration were the groove opening angle on the T40 side and the Si content in the filler material. Peyre et al. [29] reported laser welding–brazing of an AA 6061 aluminum alloy and a Ti6Al4V titanium alloy under a lap configuration without filling material and with Al-5Si filling wire. They found that the influence of Si content on mechanical strength of the joint was indistinctive. Sahul et al. [30] investigated the disk laser welding–brazing of an AW5083 aluminum alloy to a titanium grade 2 alloy with 5087 aluminum alloy filling wire. They detected that the grains in the fusion zone were refined due to the high cooling rate, and the highest tensile strength was recorded in the case of the joint whose laser beam offset was 300 μm toward the aluminum alloy plate side.

This study aimed to investigate the influences of welding parameters on dissimilar joint properties between a 5A06 alloy plate and a Ti6Al4V alloy plate. The welding–brazing connection between the Ti6Al4V alloy plate and the 5A06 alloy plate without groove and filler metals was realized by a self-designed optic fiber laser welding system under a butt configuration. The effects of welding parameters on the spreading length of the molten aluminum liquid, interface fracture zone width (IFZW), fracture roughness, and IMC thickness were analyzed. The influence factors of tensile strength of the joint are emphatically discussed.

## 2. Experimental Details

### 2.1. Material Properties

Dissimilar metals of a 5A06 aluminum plate and a Ti6Al4V titanium plate with a size of 40 mm × 60 mm × 1.5 mm were chosen for the object of study. The Ti6Al4V alloy is the α + β titanium alloy containing a certain amount of aluminum element and different amounts of the β stable elements and neutral elements. However, the 5A06 alloy is a Al–Mg series non-heat treatment strengthening alloy mainly composed of α solid solution, β phase (Al_3_Mg_2_) and β’ phase (Mg_23_Al_30_) [31]. The chemical compositions and the physical and mechanical properties of the base metals at room temperature are listed in Table 1 and Table 2, respectively.

### 2.2. Set-Up of the Welding System

The self-built fiber laser welding system equipped with a five-axis high precision sports platform driven by a servo motor was applied to connect the two specimens. A laser welding head equipped with a collimating lens with a 100 mm focusing distance and a focusing lens with a 200 mm focusing distance was utilized to deliver the laser beam with a Gaussian distribution model on the surface of the plate. Basing on the results of preliminary study, the laser defocus was adjusted to 0 mm, and the angle (β) between the shielding gas pipe and the horizontal plane was set to 30–45 degrees. In order to avoid a sharp metallurgical reaction between dissimilar metals that leads to the formation of a large number of brittle IMCs, the laser source was focused vertically (α = 90°) at a set distance from the sheets’ contact line on the side of the 5A06 plate, which is known as laser offset welding (LOW). The configuration of the welding–brazing process is shown in Figure 1.

### 2.3. Experiment Plan

Rapid heating and cooling methods are usually adopted to reduce the duration of the molten metal in liquid state to inhibit the production of IMCs. According to previous experimental results, the shielding gas flow rate was regulated to 20 L/min at the front and 15 L/min at the back. Moreover, the position of the laser beam and the supply of the heat are crucial to the laser welding experiment. Hence, laser offset, welding speed, and laser power are considered key parameters in the welding process. The welding parameters of the experiment are listed in Table 3. The experimental design has been applied to study the effects of laser offset, welding speed, and laser power on joint properties such as spreading length, IFZW, fracture roughness, IMC thickness, and tensile strength.

Before welding, the edges of the couple plates were polished until they fit closely with each other such that a weld could be formed. In addition, the oxide and oil stains on the plates were removed prior to joining together to stop slag from entering the weld. Firstly, an acetone solution was adopted to eliminate the oil stain on the plates. Secondly, the plates were polished with 80# mesh water sandpaper for a few minutes to reduce the reflectivity of the aluminum plate and preliminarily remove the oxide layer on the plates. Thirdly, a weak acid solution with 5 mL of HF, 30 mL of HNO_3_, and 65 mL of H_2_O was used to further clean the oxide layer. Finally, both the 5A06 aluminum and the Ti6Al4V titanium specimens were each cleaned with water and dried with a blow-dryer for a few minutes.

After welding, the tensile specimen and metallographic specimen as shown in Figure 1 were obtained from the weldment by a linear cutting machine. The tensile property of the joint was tested by a universal testing machine. A cross section of the metallographic specimen was first mechanically rough-polished with 180–7000# mesh water sand paper in turn, and then precision-polished via diamond spray with a particle size of 0.5 μm. Macroscopic images of the weld seam surface and cross section were obtained by optical microscope. Furthermore, the microstructure of the cross section and the fractured surfaces were observed by a scanning electron microscope (SEM, FEI Company, Hillsboro, TX, USA) equipped with an energy dispersive spectroscopy (EDS) analyzer (EDS, FEI Company, Hillsboro, TX, USA). The phase composition of the IMCs layer was observed with an X-ray diffractometer (XRD, Rigaku Rapid IIR, Japanese Electronics, Tokyo, Japan).

## 3. Weld Metallurgy and Fracture

### 3.1. Weld Structure and Composition

As shown in Figure 2a, the cross section of the joint was mainly divided into three parts: the Ti6Al4V titanium zone, the aluminum fusion zone (FZ), and the 5A06 aluminum zone. In the process of welding–brazing, some of the aluminum liquid was spread on the titanium plate. Titanium element at the Ti/FZ interface was then diffused into the FZ to form the Ti-rich crystal, and the top and bottom corners of the Ti6Al4V plate were molten, as shown in Figure 2b. According to Figure 2b, the definition of the spreading length is the maximum distance that aluminum liquid can spread over the titanium plate. The longer the spreading length is, the larger the interfacial bonding zone is, and the greater the tensile strength of the joint is. Figure 2c shows the microstructure of the cross section located at the C region in Figure 2b. The microstructure of the Ti6Al4V titanium alloy was composed of α-phase and β-phase, and the α-phase was surrounded by β-phase, which is in accordance with the description in [32]. A thin IMC layer was formed at the interface between the Ti6Al4V alloy and the FZ, and some Ti element infiltrated into the FZ to form the virgate or laminated Ti-rich compounds. Figure 2d shows the microstructure of the cross section located at the A region in Figure 2b. The EDS scanning results from P1 to P6 are listed in Table 4. The results show that the IMC layer was composed of Ti and Al compounds that can seriously damage the strength of joints, so these compounds should be as controlled as possible [33,34]. Figure 2e shows the change in Ti, Al, V, and Mg element content during scanning along the AB line in Figure 2c. It can be seen that the content of these elements varied greatly in the IMC layer. Figure 2f displays the micro-XRD pattern of the IMC layer shown in the D region of Figure 2d at the cross section of the weld. The phase of the interlayer may have consisted of solid solutions of Al and Ti as well as IMCs, such as TiAl_3_, TiAl, TiAl_2_, and Ti_3_Al.

### 3.2. Fracture Properties

Figure 3 shows the tensile specimen characteristics of the joints. The tensile specimens were designed according to the ISO standard of 6892-1:2009 [35], as displayed in Figure 3a. When the heat input reached a certain low value, only the top and bottom corners of the Ti6Al4V plate melted and formed a certain arc angle, as shown in the A and B regions of Figure 2b. If the joint ruptured all along the interface of Ti/FZ, the fracture surface would be a U-shaped surface along the interface fracture zone. When the joint was partially disrupted at the interface of Ti/FZ and the remaining part breaks in the FZ, the fracture surface would form three parts: the fracture surfaces broken at the FZ of both the topside and the backside and the fracture surface broken at the Ti/FZ interface. The interface fracture zone width (IFZW) is the width of the typical fracture surface broken along the Ti/FZ interface, as shown in Figure 3b. Because the interface fracture area is not easy to measure, only the width of the interface fracture zone was chosen to characterize the size of the area. The interface reaction in the interface fracture zone was insufficient, and a large area of non-metallurgical reaction occurred, which might have caused the connection of the region only to mechanical bonding, so the region severely reduced the overall tensile strength of the joint. The definition of fracture roughness is the difference value between the highest and lowest values of the longitudinal profile located somewhere on the fracture, as shown in Figure 3c.

Figure 4 shows the Ti6Al4V side fracture surface of Sample 5, Sample 9, and Sample 14, while the results of EDS scanning from the Z1 to Z10 regions are listed in Table 4. The content of the Al element in the Z1 and Z3 regions was especially high and the possible phase was an Al solid solution, which may have leaded to ductile fracture. The content of Ti in the Z2, Z9, and Z10 regions was relatively high, the ratio of Ti atom to Al atom was close to 3:1, and the possible phase was Ti_3_Al, which may have leaded to brittle fracture [36]. In terms of element composition, the content of Al in the Z4 and Z6 regions was also relatively high, and the possible phases were TiAl_3_ and TiAl_2_, which may have caused brittle fracture. The content of Al element in the Z5, Z7, and Z8 regions was lower than the limit solubility of aluminum in titanium according to the Ti–Al phase diagram, so the possible phase was α-Ti. Brittle fracture may have occurred in the Z7 region, while Z5 and Z8 were the interface fracture, and their connection mode was only a mechanical bite, so the bonding strength was the lowest. The possible phase of fracture is illustrated in Figure 2f. In general, the larger the fracture areas of Ti-rich, TiAl_2_, TiAl_3_, and Ti_3_Al in the FZ were, the more easily the brittle fracture occurred, and the lower the tensile strength of the joint was. On the contrary, the larger the fracture area of Al-rich in the FZ, the more easily the ductile fracture occurred, and the higher the tensile strength of the joint. The fracture surface area broken at the Ti/FZ interface was belonged to the mechanical bonding. Hence, the tensile strength of the joint increased with the enlargement of the area of interface fracture.

## 4. Discussion

### 4.1. Effects of Laser Offset

The effect of laser offset on the properties of the joint was investigated by setting the parameters of Samples 1–5 in Table 3. As shown in Figure 5a, when the laser offset was less than 300 μm, the weld seam was relatively smooth and full. As displayed in Figure 5b–d, the weld seam became unsaturated as the laser offset increased from 400 to 600 μm. The larger the laser offset, the wider the wetting and spreading distance of the molten aluminum liquid, and the insufficient weld seam naturally formed when the aluminum liquid was constant. During the welding process, spatters near the weld seam increased with the increase in laser offset [37], as shown in Figure 5a–d. Moreover, a groove formed at the bottom of the weld seam under the laser irradiation and gradually deepened with the decrease in laser offset. The smaller the laser offset, the stronger the keyhole effect, and the less molten metal fills the void left by the keyhole effect, which is consistent with the results in [26].

Figure 6a shows that the spreading length of both the topside and the backside increased first and then decreased as the laser offset increased, and the effect of laser offset on the spreading length of the backside was more significant than that of the topside. When the laser offset was relatively small, the heat loss of the molten pool decreased as the laser offset increased, which prolonged the solidification time of the molten aluminum liquid and led to an increase in the spreading length of both the topside and the backside [17,26]. However, when the laser offset exceeded 400 and 600 μm respectively, the increasing distance between the laser spot and the titanium plate led to a significant increase in heat loss on molten pool, resulting in an increase in solidification time. Therefore, the spreading length of both the topside and the backside decreased, especially the spreading length of the backside reduces to zero, as shown in the A area in Figure 6e.

Figure 6b displays that the IFZW increased with the increase in laser offset. The heat loss increased and the interface reaction weakened gradually with the increasing laser offset, resulting in a decrease in interface bonding strength [32]. Thus, the fracture occurred along the interface easily and the IFZW gradually increased. As shown in Figure 6c, when the laser offset increased from 300 to 600 μm, the fracture roughness decreased dramatically. When the laser offset was more than 600 μm, the fracture roughness tended to increase. When the laser offset increased between 300 and 600 μm, lowering the molten pool temperature from the enlargement of heat loss attenuated the fracture layer, leading to a drop in fracture roughness. However, when the laser offset exceeded 600 μm, the heat loss of the molten pool increased further, resulting in a further decline in the solidification time of the molten aluminum liquid and the further weakening of the interface reaction. Therefore, a U-shaped fracture was formed along the interface, leading to an increase in fracture roughness.

Figure 6d shows that the IMC thickness decreased with the increase in laser offset. As the laser offset increased, the increasing heat loss led to a decline in the temperature of the molten pool and a weakening of metallurgical reaction, resulting in a decrease in IMC thickness [38]. Figure 6d also shows that the tensile strength increased first and then decreased as the laser offset increased. When the laser offset increased from 300 to 600 μm, the tensile strength of the joint increased as the laser offset increased. Nevertheless, when the laser offset was up to 600 μm, the tensile strength reached a maximum value of 127 MPa. When the laser offset was more than 600 μm, the tensile strength of the joint tended to decrease. Although the IFZW (see Figure 6h) increased with laser offset varying from 300 to 600 μm, the thickness of the brittle IMCs decreased, while the spreading length of the topside and backside increased gradually. Under the condition of the lower molten pool temperature, the area of Al-rich ductile fracture with a honeycomb-like pattern in Figure 6f increased, while the area of the Ti_3_Al-rich brittle fracture in Figure 6g decreased gradually. The increasing local plastic deformation at the crack tip and the slow growing speed of the crack led to the gradual increase in the bonding strength of the joint [27,30]. When the laser offset exceeded 600 μm, although the thickness of the brittle IMCs decreased dramatically, the IFZW increased and the spreading length of the topside and backside decreased remarkably, resulting in a sharp decrease in the area of the fracture broken in FZ. Therefore, the tensile strength decreased sharply under such a condition.

### 4.2. Effects of Welding Speed

The influence of welding speed on properties of the joint was explored by setting parameters of Samples 6–9 in Table 3. As shown in Figure 7a, when the welding speed was 8 mm/s, the weld surface was relatively smooth and a large hole formed in the left side of the weld. This is because when the welding process started, almost all laser energy was used to melt the metal. Therefore, the heat loss was slight, and the material was overheated and burned. As the welding process continued, heat dissipation increased, and the rest of the energy could only melt metal to form welds. Spatters increased with the increase in welding speed during the welding process, as presented in Figure 7b–d. In addition, a groove formed at the bottom of the weld seam under the laser irradiation and gradually deepened as the welding speed decreased. This is because the lower the welding speed was, the stronger the keyhole effect was, and the less molten metal fills the void left by the keyhole effect [26].

Figure 8a shows that the spreading length of both the topside and the backside decreased as the welding speed increased, and the influence of welding speed on the spreading length of the backside was more significant than that of the topside. When the welding speed increased, the heat on the unit area decreased, and there was not enough molten aluminum liquid to spread on the titanium plate, leading to a decrease in spreading length; the spreading length of the backside especially reduced to zero, as shown in the A area in Figure 6e.

Figure 8b shows that the IFZW increased as the welding speed increased. The temperature of the molten pool decreased dramatically and the heating time of the molten pool reduced sharply as the welding speed increased, leading to a weakening of the interface reaction and a decrease in interface bonding strength. As a result, the fracture occurred more easily along the interface, so the IFZW increased gradually. As displayed in Figure 8c, the fracture roughness also decreased as the welding speed increased in a small range. This is because the decrease in the heat per unit area led to a weakening of the interface reaction and a thinning of the IMC fracture layer, resulting in a reduction in fracture roughness.

Figure 8d shows that the IMC thickness decreased as the welding speed increased. The temperature and heating time of the molten pool decreased as the welding speed increased, resulting in a decrease in the IMC thickness. However, the spreading length of the topside and backside decreased and the IFZW (see Figure 8f) increased as the welding speed increased, which made the joint more likely to break. However, the tensile strength increased with the increase in welding speed, as shown in Figure 8d. This is because the decline in molten pool temperature may cause a weakening of the interface reaction, a sharp decrease in the IMCs, an increase in the area of Al-rich ductile fracture in Figure 6f, and a decrease in the area of the TiAl_3_-rich brittle fracture in Figure 8e. Under this condition, the local plastic deformation increases at the crack tip and the crack grows slowly, which leads to an increase in bonding strength and the tensile strength of the joint [39]. When the welding speed was up to 11 mm/s, the tensile strength reached 183 MPa, which is 83% of the strength of the 5A06 aluminum alloy base metal.

### 4.3. Effects of Laser Power

The influence of laser power on the properties of the joint was explored by setting parameters of Samples 10–14 in Table 3. During the welding process, spatter occurs near the weld seam and increased with the increase in laser power, as shown in Figure 9. As presented in Figure 9a, when the laser power was less than 1080 W, the weld seam was relatively narrow and not full. As the laser power increased from 1130 to 1230 W, the weld seam became wider and fuller, as displayed in Figure 9b–d. This is because the larger the laser power, the larger the size of the molten pool, the stronger the wetting and spreading ability, and the wider and fuller the weld seam naturally is [18,26]. Moreover, a groove formed at the bottom of the weld seam under the laser irradiation and gradually deepened as the laser power increased, as shown in Figure 9. This is because the higher the laser power is, the stronger the keyhole effect is, and the less molten metal fills the void left by the keyhole effect [26].

Figure 10a shows that the spreading length of both the topside and the backside increased first and then decreased as the laser power increased, and the spreading length of the backside was greater than the topside. When the laser power was less than 1230 W, the spreading length of both the topside and the backside increased to 1471 and 1602 μm at 1230 W, respectively. When the laser power was greater than 1230 W, the spreading length of both the topside and the backside dropped sharply, but the top corner of the Ti6Al4V sheet melted more, as shown in the A area in Figure 10e. When the laser power is small, there is not enough molten aluminum liquid to spread over the Ti6Al4V titanium alloy plate under given process parameters [26,31]. As the laser power increased, the spreading length of the molten aluminum liquid was higher because of the increasing molten aluminum liquid. However, when the laser power exceeded 1230 W, more heat diffused into the air or participated in the interface reaction under given parameters, resulting in the rapid solidification of molten aluminum liquid and a sharp decrease in spreading length.

Figure 10b shows that the IFZW decreased as the laser power increased. This is because the increase in the molten pool temperature and the decrease in the heat loss led to a gradual strengthening of the interface reaction and a slow strengthening of the interface bonding strength. Under this condition, the fracture could not easily occur along the interface, so the IFZW decreased gradually. The effect of laser power on fracture roughness is shown in Figure 10c. When the laser power increased from 1080 to 1230 W, the fracture roughness decreased slowly. When the laser power increased to more than 1230 W, the fracture roughness increased rapidly. The main reason is that, when the laser power is small, heat effect is relatively insignificant and the interface reaction is not obvious [18]. Hence, the intermediate layer was very weak and easy to fracture. In addition, the fracture surface was not a plane but a curved surface (see Figure 3c), so the fracture roughness was relatively large. As the laser power increased, the interface reaction became more significant, and the fracture surface gradually moved toward the FZ rather than just at the interface. Therefore, the arc of the fracture surface was smaller and tended to be flat, and the fracture roughness decreased gradually. When the laser power was more than 1230 W, excessive heat was involved in the interface reaction, leading to the formation of thicker IMCs, so the fracture roughness increased sharply.

Figure 10d shows that the IMC thickness increased as the laser power increased, which is similar to the results in [27]. This is because the molten pool temperature increased as the laser power increased and the metallurgical reaction easily occurred, so the IMC thickness increased as the laser power increased [33]. Figure 10d also shows that the tensile strength of the joint increased first and then decreased as the laser power increased. Although the IMC thickness increased gradually as the laser power increased from 1080 to 1230 W, the joint was more likely to break. However, the IFZW (see Figure 10f) decreased and the spreading length of the topside and backside increased as the laser power increased. Hence, the tensile strength of the joint increased as the laser power increased [40,41]. In addition, although the area of the Ti_3_Al-rich brittle fracture in Figure 10g,h was relatively large, the crack propagation was fast under such heating condition. Therefore, the strength of the joint increased gradually with the sharp decrease in IFZW. When the laser power was 1230 W, the tensile strength of the joint reached 133 MPa. When the laser power was more than 1230 W, although the IFZW continued to decrease, the joint could not easily break. However, the IMC thickness increased sharply, the area of the brittle fracture in FZ continued to increase, and the cracks expanded rapidly, which deteriorated the tensile property of the joint. Hence, the tensile strength was decreased under such a condition.

## 5. Conclusions

The laser welding–brazing of Ti6Al4V and 5A06 plates with a 1.5 mm thickness in a butt configuration was carried out by focusing a laser beam on a 5A06 aluminum alloy side without a filler metal. The effects of laser offset, welding speed, and laser power on the spreading length, IFZW, fracture roughness, IMC thickness, and tensile strength of the joint were investigated. The conclusions are as follows:
The Ti6Al4V titanium plate and 5A06 aluminum plate were successfully jointed by a laser welding–brazing process. A dissimilar butt joint with an IMC thickness of 2.79 μm was obtained by adjusting the laser offset, welding speed, and laser power to 500 μm, 11 mm/s, and 1130W, respectively. The maximum tensile strength of the joint was up to 183 MPa, which is equivalent to 83% of the tensile strength of the 5A06 aluminum alloy.The spreading length of both the topside and the backside increased first, then decreased with the increase in laser offset and laser power, and decreased with the increase in welding speed. The IFZW increased with the increase in laser offset and welding speed, and decreased with the increase in laser power. The fracture roughness decreased first, then increased with the increase in laser offset and laser power, and decreased with the increase in welding speed. The IMC thickness decreased with the increase in laser offset and welding speed, and increased with the increase in laser power.The tensile strength of the joint was influenced not only by the thickness and type of IMC, but also by the spreading ability of the aluminum liquid, the fracture area broken at the Ti/FZ interface, the relative area of the brittle, and the ductile fracture in FZ.


## Figures and Tables

**Figure 1 materials-12-00712-f001:**
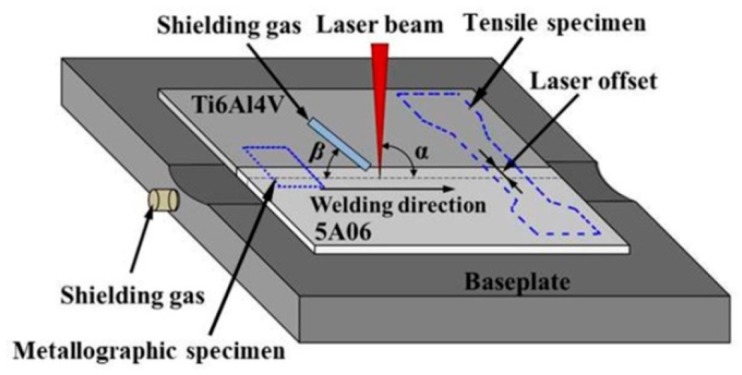
Schematic diagram of welding process.

**Figure 2 materials-12-00712-f002:**
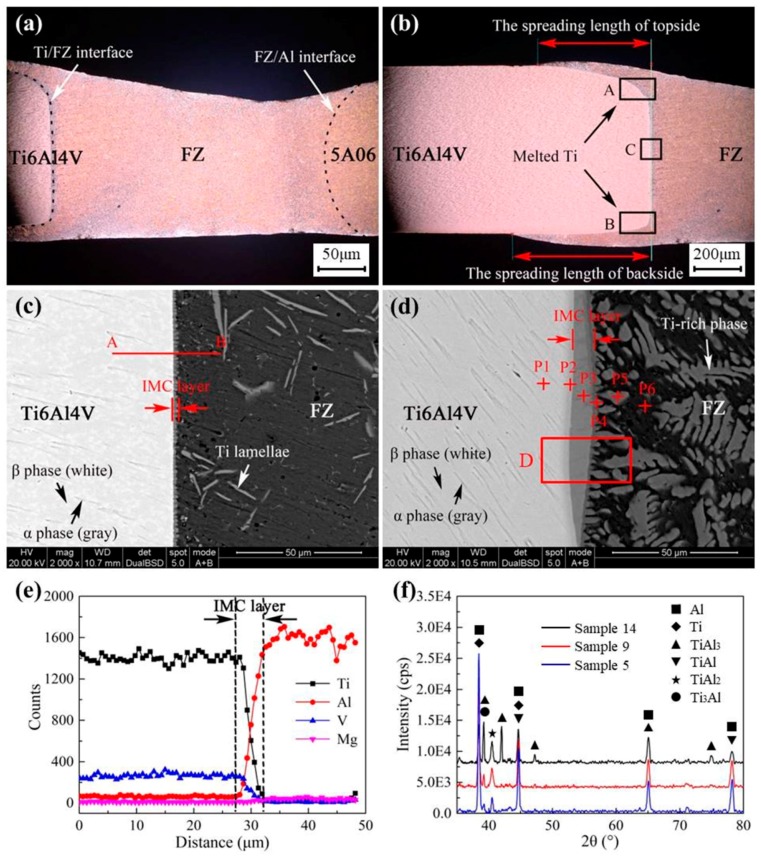
Cross section of the joint: (**a**) macro cross section; (**b**) local graph of cross section; (**c**) SEM image of the C region; (**d**) SEM image of the A region; (**e**) EDS line scanning along AB; (**f**) XRD scanning results of the D region in Figure 2d.

**Figure 3 materials-12-00712-f003:**
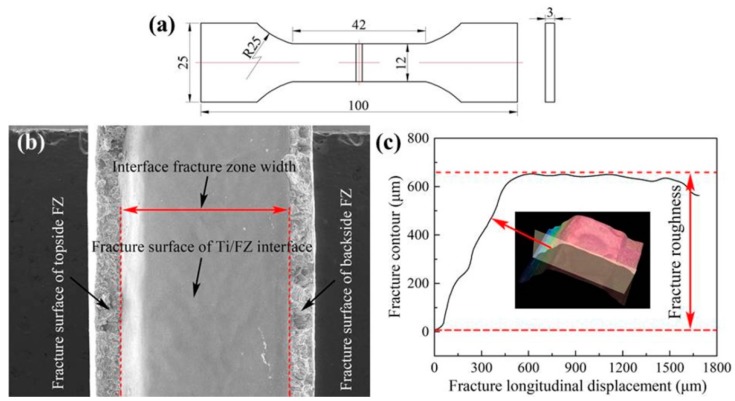
Tensile specimen characteristics: (**a**) 2D drawing; (**b**) fracture surface; (**c**) definition of fracture roughness.

**Figure 4 materials-12-00712-f004:**
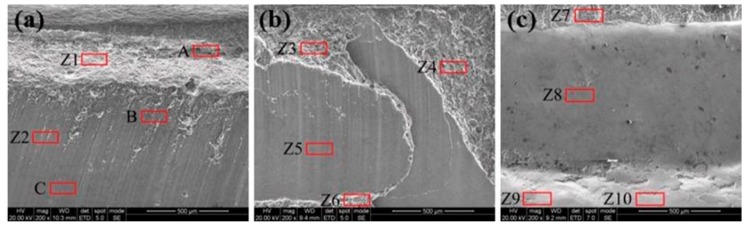
SEM images of fracture: (**a**) Joint #5; (**b**) Joint #9; (**c**) Joint #14.

**Figure 5 materials-12-00712-f005:**
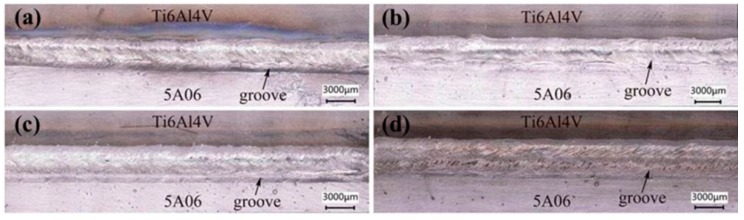
Influence of laser offset on weld seam appearance: (**a**) 300 μm; (**b**) 400 μm; (**c**) 500 μm; (**d**) 600 μm.

**Figure 6 materials-12-00712-f006:**
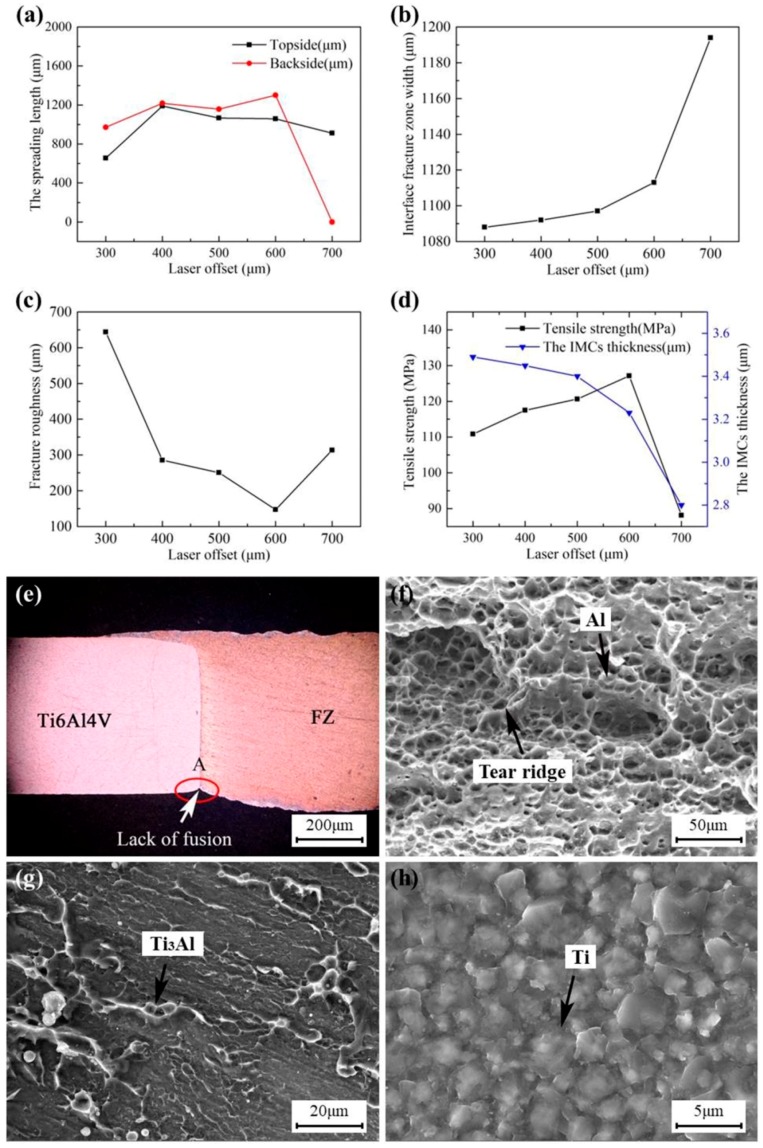
Influence of laser offset on the joint properties: (**a**) the spreading length; (**b**) interface fracture zone width; (**c**) fracture roughness; (**d**) tensile strength and the IMC thickness, and cross section and details of the fracture of Joint #5: (**e**) cross section of the joint; (**f**–**h**) details of Areas A, B, and C in Figure 4a, respectively.

**Figure 7 materials-12-00712-f007:**
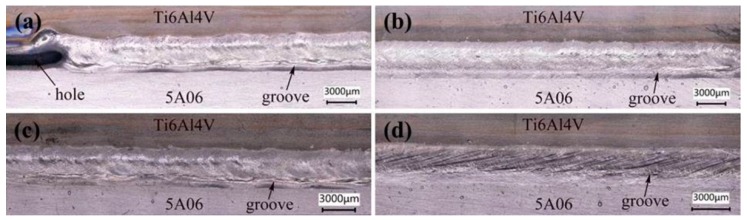
The influence of the welding speed on weld seam appearance: (**a**) 8 mm/s; (**b**) 9 mm/s; (**c**) 10 mm/s; (**d**) 11 mm/s.

**Figure 8 materials-12-00712-f008:**
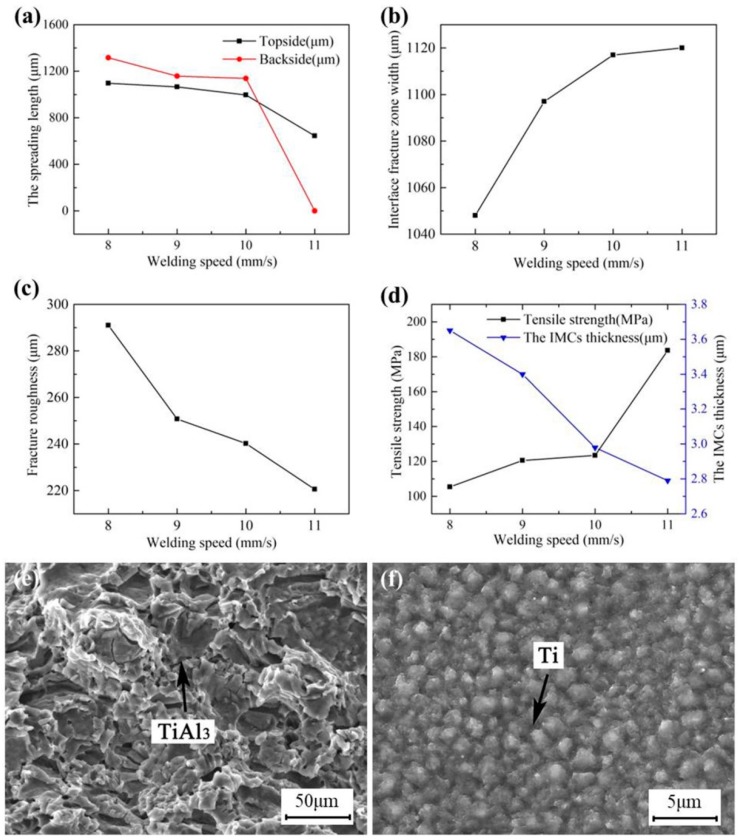
Influence of welding speed on the joint properties: (**a**) the spreading length; (**b**) interface fracture zone width; (**c**) fracture roughness; (**d**) tensile strength and the IMC thickness, and the details of the fracture of Joint #9: (**e**,**f**) details of Areas Z4 and Z5 in Figure 4b, respectively.

**Figure 9 materials-12-00712-f009:**
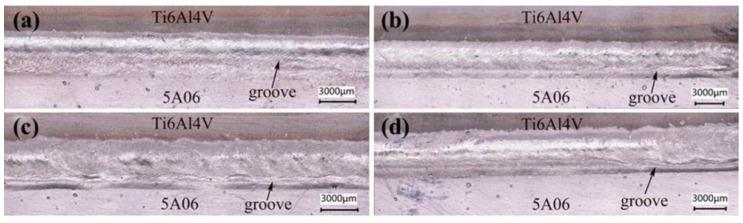
Influence of laser power on weld seam appearance: (**a**) 1080 W; (**b**) 1130 W; (**c**) 1180 W; (**d**) 1230 W.

**Figure 10 materials-12-00712-f010:**
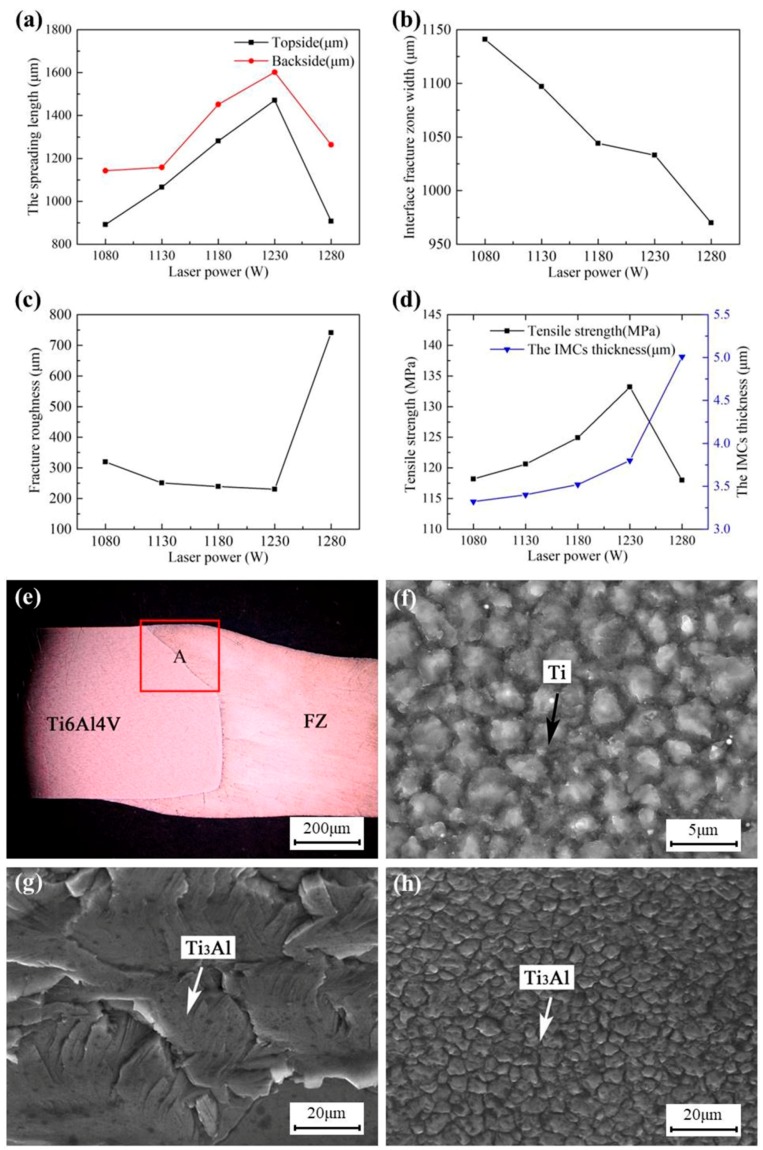
Influence of laser power on the joint properties: (**a**) the spreading length; (**b**) interface fracture zone width; (**c**) fracture roughness; (**d**) tensile strength and the IMC thickness, and cross section and details of the fracture of Joint #14: (**e**) cross section of the joint; (**f**–**h**) details of Areas Z8, Z9, and Z10 in Figure 4c, respectively.

**Table 1 materials-12-00712-t001:** Chemical composition of the 5A06 and Ti6Al4V alloys.

Alloy	Elements (wt.%)												
	Al	Ti	Mg	Si	Cu	Mn	Fe	Zn	V	C	N	H	O
5A06	Bal.	0.02	5.8–6.8	0.4	0.1	0.5–0.8	0.4	0.2	–	–	–	–	–
Ti6Al4V	5.5–6.8	Bal.	–	–	–	–	0.3	–	3.5–4.5	0.1	0.05	0.01	0.2

**Table 2 materials-12-00712-t002:** Physical and mechanical properties of base materials: density (D), thermal conductivity (TC), coefficient of linear expansion (CLE), tensile strength (TS), yield strength (YS), and elasticity modulus (EM).

Alloy	Density (g/cm^3^)	Thermal Conductivity (W/(m·K))	Coefficient of Linear Expansion (K^−1^)	Tensile Strength (MPa)	Yield Strength (MPa)	Elasticity Modulus (GPa)
5A06	2.64	117	24.7 × 10^−6^	325	160	68
Ti6Al4V	4.44	7.95	8.6 × 10^−6^	967	860	112

**Table 3 materials-12-00712-t003:** The process parameters.

Sample	Laser Power (W)	Welding Speed (mm/s)	Laser Offset (μm)	Shielding Gas Flow Rate (L/min)	
				Top	Back
1	1130	9	300	20	15
2	1130	9	400	20	15
3	1130	9	500	20	15
4	1130	9	600	20	15
5	1130	9	700	20	15
6	1130	8	500	20	15
7	1130	9	500	20	15
8	1130	10	500	20	15
9	1130	11	500	20	15
10	1080	9	500	20	15
11	1130	9	500	20	15
12	1180	9	500	20	15
13	1230	9	500	20	15
14	1280	9	500	20	15

**Table 4 materials-12-00712-t004:** EDS results of P1–P6 points in Figure 2d and Z1–Z10 regions in Figure 4.

Zones	Chemical Composition (at%)				Probable Phase
	Ti	Al	V	Mg	
P1	90.01	5.51	3.98	0.50	Ti
P2	73.70	22.40	3.37	0.53	Ti_3_Al
P3	52.09	44.64	2.87	0.39	Ti_3_Al + TiAl
P4	36.34	61.42	1.17	1.06	TiAl + TiAl_2_
P5	29.99	66.79	1.59	1.63	TiAl_2_
P6	3.58	93.67	0.56	2.18	Al
Z1	2.38	94.45	0.35	2.82	Al
Z2	63.89	32.14	2.63	1.34	Ti_3_Al
Z3	5.20	91.60	0.50	2.70	Al
Z4	11.38	83.76	2.06	2.81	TiAl_3_
Z5	83.25	10.88	5.12	0.75	Ti
Z6	27.49	67.41	3.05	2.05	TiAl_2_
Z7	82.14	10.76	6.39	0.72	Ti
Z8	86.44	8.44	4.85	0.28	Ti
Z9	76.39	19.59	3.70	0.33	Ti + Ti_3_Al
Z10	63.28	31.61	3.68	1.42	Ti_3_Al

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
