# Peer review of "The Study on Mechanical Strength of Titanium-Aluminum Dissimilar Butt Joints by Laser Welding-Brazing Process"

_materials, 2019, doi:10.3390/ma12050712_

Reviewer 1 Report

Abstract and introduction - the purpose of the article should be clearly written

Table 2 - shortcuts should be replaced with full names

The graphs, figures are made very carefully.

The work is interesting. It does not raise any reservations.

Author Response

Dear reviewer,

Thank you for your valuable time in reviewing my paper. I'm glad to receive your suggestion for amendment. In accordance with your suggestion, we have made the corresponding modification of the paper, and now we will send the result of the modification to you in the form of a cover letter. Please consult it in time.

Best regards,

Xiongfeng Zhou

Reviewer 2 Report

The article is well-written describing the effect of the operational parameters of laser welding-brazing on joining of two dissimilar metals, aluminum alloy, and titanium alloy. The mechanical efficiency of the butt joint has been investigated over the tensile strength with respect to that of the original metal. This article experimentally illustrates how operational parameters can influence the significant features of the butt-joints and in turn can affect the mechanical strength of the joint.

Upon a careful review of this article, the following comments have been extended -

(1) Minor typographical errors throughout the article need correction. Example, Line 337.

(2) Figure 10. a-d have Laser Power (W) axes with wrong power values, need to be corrected.

(3) The results need to supported by statistical significance. Only one test for each configuration merely gives an incidental observation. A conclusion about a system needs multiple tests and statistical analysis.  

Author Response

(The authors gave the same response as above.)

Reviewer 3 Report

The work discussed the subject The study on mechanical strength of titanium-aluminium dissimilar butt joint by laser welding-brazing process. Presented research stays interesting, in the reviewer opinion, however, stays more close to a technical report nether research article. If, however, the editor decides to consider the submission, the quality of work should be improved before publishing. At the moment I advise against the publishing or major revision of the text.

General

1.     From the beginning, authors should use proper wording IMG’s is a zone or area, leaving it alone at the description may introduce interpretation misunderstandings. L(14)

2.     What does it mean “harmoniously fused”? L(36)

3.     It is unclear from the beginning, why the authors prepare the samples in the specific processing range Table 3, the intention should be explained at the experimental plan subsection. For the design of the experiment, and more reliable data (new experimental matrix) other broadly describe in literature methods could be useful for example Taguchi one. For above collate data it remains also unclear for the reviewer, why the authors include not changeable parameters in the table, as also why to repeat one configuration three times sample 3-7-11. Is there some results scatter, the authors did not mention anything about it, as also do not show any of the results. L(114)

4.     The third step of the sample preparation procedure, seems to be unnecessary after etching solution agent treatment? Or the sequence remains in not proper order for the oil stains removing? L(121)

5.     Are the wording “graph weld” is correctly used? L(126)

6.     For the XRD analysis, what region of the samples the diffractograms presents? How the samples were prepared for analysis? L(129)

7.     Are the wording “illuminated” is correctly used? L(136)

8.     Why the phase composition of the base plate of Ti6Al4V is referenced nether outcome from the microstructure description of metallographic specimens? L(141)

9.     For the weld structure composition at the interlayer section, used by the author's examples (5,9,14) should be in the same way described. It remains unclear why the authors collate those samples for comparison. L(149)

10.  “small” L(337)

Author Response

(The authors gave the same response as above.)
